# SARS-CoV-2 Surveillance between 2020 and 2021 of All Mammalian Species in Two Flemish Zoos (Antwerp Zoo and Planckendael Zoo)

**DOI:** 10.3390/vetsci10060382

**Published:** 2023-05-31

**Authors:** Léa Joffrin, Tine Cooreman, Erik Verheyen, Francis Vercammen, Joachim Mariën, Herwig Leirs, Sophie Gryseels

**Affiliations:** 1Evolutionary Ecology Group, Department of Biology, University of Antwerp, 2610 Antwerp, Belgium; 2OD Taxonomy and Phylogeny, Royal Belgian Institute of Natural Sciences, 1000 Brussels, Belgium; 3Centre for Research and Conservation, Antwerp Zoo Society, 2018 Antwerp, Belgium

**Keywords:** SARS-CoV-2, surveillance, zoo, mammals, Belgium

## Abstract

**Simple Summary:**

COVID-19 emerged in China in 2019. It is caused by an until-then unknown coronavirus that causes severe acute respiratory syndrome (SARS-CoV-2). Through experimental infections in the search for a suitable animal model and reported infections in pets early in the pandemic, it became clear that several animal species may be susceptible to SARS-CoV-2 infection. According to the open access dataset of reported SARS-CoV-2 events in animals, about 119 zoo animals have been reported with a SARS-CoV-2 infection. However, the detection of SARS-CoV-2 infections in zoo animals has relied on the observation of symptoms (cough, nasal discharge), behaviour changes (reduced appetite, lethargy), or death of these captive animals. SARS-CoV-2 infections may therefore remain undetected if animals do not show obvious symptoms. In this study, we investigated the potential circulation of SARS-CoV-2 in zoo mammal species by sampling and screening faecal samples from all the mammals in two zoos in Belgium between September 2020 and July 2021 using molecular biology techniques. This study is the first to our knowledge to conduct active SARS-CoV-2 surveillance for several months in all mammal species in a zoo. We conclude that at the time of our investigation, none of the screened animals were excreting SARS-CoV-2.

**Abstract:**

The COVID-19 pandemic has led to millions of human infections and deaths worldwide. Several other mammal species are also susceptible to SARS-CoV-2, and multiple instances of transmission from humans to pets, farmed mink, wildlife and zoo animals have been recorded. We conducted a systematic surveillance of SARS-CoV-2 in all mammal species in two zoos in Belgium between September and December 2020 and July 2021, in four sessions, and a targeted surveillance of selected mammal enclosures following SARS-CoV-2 infection in hippopotamuses in December 2021. A total of 1523 faecal samples from 103 mammal species were tested for SARS-CoV-2 via real-time PCR. None of the samples tested positive for SARS-CoV-2. Additional surrogate virus neutralisation tests conducted on 50 routinely collected serum samples from 26 mammal species were all negative. This study is the first to our knowledge to conduct active SARS-CoV-2 surveillance for several months in all mammal species of a zoo. We conclude that at the time of our investigation, none of the screened animals were excreting SARS-CoV-2.

## 1. Introduction

COVID-19 emerged in China in 2019. It is caused by an until-then unknown coronavirus that causes severe acute respiratory syndrome (SARS-CoV-2). This infectious disease spread to all continents in a few months and was declared a pandemic by the World Health Organisation in March 2020. SARS-CoV-2 can be transmitted through three main routes: direct contact with infected secretions (saliva, respiratory secretions), droplet transmission (when coughing or sneezing), and aerosol transmission [1]. Through experimental infections in the search for a suitable animal model and reported infections in pets early in the pandemic, it became clear that several animal species may be susceptible to SARS-CoV-2 infection [2,3,4,5,6,7,8]. Experimental in vivo and in vitro infections showed that SARS-CoV-2 can infect a broad taxonomic range of mammals, including a.o. North American deer mice (*Peromyscus maniculatus*), macaques (*Macaca mulatta* and *Macaca fascicularis*), domestic cats (*Felis catus*), ferrets (*Mustela putorius furo*), American mink (*Neovison vison*), raccoon dogs (*Nyctereutes procyonoides*), Syrian hamsters (*Mesocricetus auratus*), and Egyptian fruit bats (*Rousettus aegyptiacus*) [3,9,10,11,12,13,14,15,16,17,18]. Circulation of SARS-CoV-2 was reported in farmed American mink (*Neovison vison*) in a multitude of farms around the world, and wild white-tailed deer (*Odocoileus virginianus*) across North America [2,3,4,5,6].

In addition, functional, structural, and genetic analysis of viral receptor ACE2 orthologs reveals that many other species may be susceptible to SARS-CoV-2 [19,20]. While these studies may appear helpful to estimate the potential host range of SARS-CoV-2, the observed natural infections highlight that susceptibility based on the ACE2 receptor alone is not a sufficient proxy to estimate potential spillover risk to other species [21]. For example, mink or wild white-tailed deer are not considered highly susceptible based on these in-silico analyses [5].

According to the open access dataset of reported SARS-CoV-2 events in animals (data from January 2023), about 119 zoo animals have been reported with a SARS-CoV-2 infection, representing 64 reported events and 17 species in 17 countries [22]. The most frequently reported infected mammals in zoos are felines, followed by primates [23]. However, the detection of SARS-CoV-2 infections in zoo animals has relied on the observation of symptoms (cough, nasal discharge), behaviour changes (reduced appetite, lethargy), or death in these captive animals [24,25]. SARS-CoV-2 infections may therefore remain undetected if animals do not show obvious symptoms. 

Since infected animals have been found in zoos worldwide, and given the long-term high incidence of the virus in humans, we deemed it prudent to monitor the presence of SARS-CoV-2 in zoo animals. Furthermore, the high diversity of zoo animals, both regarding taxonomy and geographical origin, makes zoos an ideal place to (i) contribute to unravelling the potential host range of SARS-CoV-2 and (ii) evaluate the risk for the conservation of wild animal populations in captivity and in situ. For this study, we investigated the potential circulation of SARS-CoV-2 in zoo mammal species by sampling and screening faecal samples from all the mammals in two zoos in Belgium in four sessions between September 2020 and July 2021, via real-time polymerase chain reactions (PCR). Following symptomatic SARS-CoV-2 infection in hippos in the Antwerp Zoo in December 2021 [26], we additionally surveyed selected mammals deemed to be in potential indirect contact with the hippos, or with expected relatively high SARS-CoV-2 susceptibility.

## 2. Materials and Methods

### 2.1. Samples Collection

We conducted this study at the Antwerp Zoo and Planckendael Zoo in, respectively, Antwerp and Mechelen, Belgium. For the systematic surveillance, we collected the samples during four periods (early September 2020, mid-October 2020, mid-December 2020 and July 2021), with sampling following enclosure cleaning planning. During the first sampling period, both zoos were still open to the public; during the second sampling series, both were closed to the public and remained closed until after the third sampling due to government regulations. The zoos reopened in February 2021, and the fourth sampling session was conducted in July 2021. During the first three sampling sessions, the original Wuhan-Hu1 variant was dominant in the human population in Belgium; during the fourth, the delta variant, considered more contagious than the previous alpha, beta, and gamma variants [27], was dominant in Belgium. Faecal samples were collected by zookeepers in a 16.5 mL tube filled with RNAlater and then stored at −20 °C at the zoo for a few days before transport to the lab, where the samples were stored at −80 °C. RNAlater is a suitable conservation medium widely used for microbiological studies [28,29]. The date and freshness of each sample were documented (maximum two hours old, or no more than twelve hours old), after which the samples were stored. A maximum of five samples per species per zoo were collected at each sampling session. A total of 1417 faeces samples were collected from 103 different mammal species (Antwerp *n* = 48 and Planckendael *n* = 67) (Table 1). In Antwerp Zoo, the largest sampled taxonomic group was the Primates, followed by the order of the Cetartiodactyla. In Planckendael, Cetartiodactyla was sampled most often, followed by the order of the Carnivora.

After our systematic surveillance, two female hippopotamuses in Antwerp Zoo showed evidence of nasal discharge in late November 2021 for a few days [26,30]. SARS-CoV-2 was detected by immunocytochemistry in nasal swab samples and by PCR in nasal swab samples, faeces, and pool water [26]. Serological tests also detected antibodies against SARS-CoV-2. 

Following these hippo infections, we conducted a targeted surveillance for SARS-CoV-2 in December 2021, collecting samples from mammals that could have been in indirect contact with the hippo individuals (i.e., if they were managed by the same caretakers) or that were of special interest due to their known increased susceptibility and conservation status, namely, primates and large felines. We screened these samples with the CDC 2019-Novel Coronavirus (2019-nCoV) Real-Time RT-PCR Diagnostic Panel, specifically targeting SARS-CoV-2 genes, and this was also used for the diagnosis of SARS-CoV-2 in the infected hippopotamuses’ faecal samples [31]. Details on the 106 samples collected and tested in December 2021 are also available in the table.

### 2.2. Sample Preparation, Extraction and PCR Testing

After thawing, the samples were processed under a Biosafety cabinet class II. Around 1 cm^3^ of the faeces was cut off, rinsed with 200 μL of phosphate-buffered saline (PBS), and mixed in a 1.5 mL Eppendorf tube filled with 800 μL of PBS. The tubes were briefly vortexed and centrifuged (1500 g for 15 min), and for each collection date/enclosure/species, samples were pooled to extract faecal RNA using the QIAGEN QIAamp viral RNA kit (Qiagen, Valencia, CA, USA) following the manufacturer recommendations. Overall, 420 pools were extracted. Reverse transcription was performed on 8 μL of RNA extract using the Maxima Reverse Transcriptase and Random Hexamer Primers (Thermo Fisher Scientific, Waltham, MA, USA) on a Biometra T3000 thermocycler (Biometra, Westburg, The Netherlands). A pan-coronavirus system suitable for the detection of alpha-, beta-, gamma- and delta-CoVs real-time PCR adapted version of the Muradrasoli et al. (2009) protocol [32,33] was used on a StepOne™ Real-Time PCR System (Applied Biosystems, Carlsbad, CA, USA) to screen the samples for all potential coronaviruses that may occur in zoo animals.

### 2.3. Validation of the PCR System for the Detection of SARS-CoV-2

We conducted assays to validate the use of the Pan-CoV system for the detection of SARS-CoV-2 in our samples. We compared the limit of detection of the Pan-CoV system targeting the RNA-dependent RNA-polymerase (RdRp) gene to the CDC 2019-nCoV Real-Time RT-PCR Diagnostic Panel, specifically targeting the SARS-CoV-2 nucleocapsid (N) gene in two multiplex reactions (N1 and N2) [31]. The limit of detection was determined to be the lowest dilution that still resulted with a Ct value. RNA from a SARS-CoV-2-positive clinical sample was used to conduct this assay. The CDC Real-Time RT PCR was performed on a serial dilution of the positive sample RNA ranging from 10^−1^ to 10^−8^. The same 8-fold dilution series was reverse-transcribed to cDNA (using the Maxima RT protocol described above), which was then run on the Pan-CoV Real-Time PCR.

In addition, a synthetic N1 and N2 gene positive control (2019-nCoV_N_Positive Control, Integrated DNA Technologies) with a known copy number was used in the CDC panel at three concentrations: 2000 copies/µL, 200 copies/µL and 20 copies/µL alongside the dilution series of the clinical sample. The Pan-CoV Real-Time PCR was used on the cDNA synthesised following the reverse-transcription protocol described above from the SARS-CoV-2-positive clinical sample RNA ranging from 10^−1^ to 10^−8^. Each dilution was tested in triplicates. 

For each system, the standard curve of the positive clinical sample was calculated by plotting the PCR cycle threshold (Ct) to the dilution number of the positive clinical sample, from which the logarithmic function (y = −a ln(x) + b) was calculated. If the R^2^ value was less than 0.96, all serially diluted RNA and cDNA was remade and retested. 

The copy number concentration of SARS-CoV-2 N gene RNA in the SARS-CoV-2 clinical sample was inferred via the standard curve of the 2019-nCoV_N_Positive Control dilution series used in the CDC system.

### 2.4. Serological Screening

Additionally, 50 blood samples from 26 mammal species were available from routine collection by the zoo veterinary service, both before (14 samples/12 species) and after (36 samples/26 species) 2020, for animals that either moved between zoos or for those requiring a veterinary follow-up (pregnancy, injury, illness). Serum samples were tested for the presence of antibodies against SARS-CoV-2 with the L00847 surrogate virus neutralisation test (sVNT) (GenScript cPass™, Piscataway, NJ, USA) as described in Mariën et al. [34]. The percentage inhibition was calculated as: ((1−OD value of sample)/OD value of Negative control) × 100%. If inhibition values were greater than 20%, serum samples were considered SARS-CoV-2-positive. Two negative serum samples, two positive serum samples and two positive serum samples from SARS-CoV-2-infected humans were used as controls. Details on the samples tested are available in Appendix A.

## 3. Results and Discussion

None of the 1523 faecal samples across the five collection periods tested positive with the pan-coronavirus screening system. Furthermore, all serum samples were seronegative for neutralising antibodies, suggesting that the tested mammals had not experienced SARS-CoV-2 infection at the time of sample collection (Appendix A). As such, apart from the infection in two hippos in December 2021 that was discovered because of clinical symptoms and not through our active surveillance study, there was no evidence of SARS-CoV-2 or other coronavirus infection among the mammals residing in the Antwerp and Planckendael Zoos during the period of the study. 

We compared the Pan-CoV PCR system used to test faecal samples collected during the first four sampling sessions between September 2020 and July 2021 with a golden standard test for SARS-CoV-2 detection (CDC N1/N2), to ensure that the sensitivity of the detection system was not an issue. We inferred the copy number per µL of a positive control SARS-CoV-2 RNA from a sample by comparing it with the known copy numbers of the N1 synthetic control of the CDC SARS-CoV-2 system. In both systems, the template was detectable up to a 10^−5^ dilution, corresponding to 2.42 N1-gene-copies/µL with Ct values of 34.76 ± 0.12 (CDC) and 38.84 ± 2.48 (Pan-CoV) (Appendix A). Hence, the detection limit and sensitivity of the CDC and the Pan-CoV system were very comparable, making it unlikely that the choice of a pan-coronavirus RT-PCR system instead of a SARS-CoV-2-specific detection system caused false-negative results. The advantage of the Pan-CoV system is that we could also determine the possible presence of other coronaviruses with one PCR test. While the entire range of coronaviruses may not be detected with the same sensitivity with this Pan-CoV system, it has been validated to detect the SARS-CoV-2 RNA-dependent RNA-polymerase (RdRp) gene.

Virus survival or the successful detection of viral RNA depends on the virus variant, the medium in which it is present and environmental conditions (temperature, pH, moisture content, organic matter, light, etc.) [35]. Although the SARS-CoV-2 virus is stable on most indoor surfaces [36,37,38], other factors in outdoor environments may reduce its survival [39]. Studies on the effect of temperature on SARS-CoV-2 survival showed that it might survive from 5 to 10 days at 20 °C and from 1 to 4 days at 30 °C, depending on the surface type [40]. Even if no studies on SARS-CoV-2 stability in faeces in outdoor environments have been conducted, a comprehensive study on the survival of several other coronaviruses in faeces concluded that SARS-CoV-2 could survive from 1 h to 4 days in human faeces, depending on the type and pH of the stool samples [35]. In our study, the delay between excretion and collection, and other environmental factors, might have influenced the quality of the samples. However, we tried to limit these issues by collecting samples that were as fresh as possible (less than 12 h after excretion). Finally, the mean temperature ranged from 0 to 22 °C during the whole sampling campaign. We therefore assume that the impact of temperature on the preservation of faeces on the ground of the enclosure will be minimal.

The non-detection of SARS-CoV-2 RNA in this study might be related to the study sampling design. Faecal samples are suitable materials for the detection of SARS-CoV-2 RNA, even if there is no consensus about which sample type (i.e., nasopharyngeal swabs, oropharyngeal swabs, faeces, or rectal swabs) is best suited to detect SARS-CoV-2 RNA, especially in non-human animals [41,42,43]. Moreover, we cannot exclude that we missed a potential SARS-CoV-2 infection in zoo mammals in our study because of the duration of SARS-CoV-2 RNA in faeces after the acute infection. Zhang et al. (2021) conducted a systematic review and meta-analysis on 14 studies on the faecal shedding of SARS-CoV-2 RNA in human patients (*n* = 620) with COVID-19 infection [42]. On average, viral RNA could be detected up to 21.8 days after infection, while nasopharyngeal swabs could only detect RNA 14.7 days after infection. The sampling sessions were, on average, six weeks apart, with over six months between the two subsequent sessions. We therefore cannot exclude that SARS-CoV-2 infections occurred between the sampled sessions. However, due to logistical reasons, more frequent sampling was not feasible. Nevertheless, longitudinal faecal screening of infected tigers and lions in the USA and hippos in Belgium showed that SARS-CoV-2 RNA could be detected up to 35 days after symptom onset [25,26]. Viral RNA shedding in these animals’ faeces may be more apparent than in humans, where only about half of the patients have detectable SARS-CoV-2 RNA in faeces at any point during infection [42]. If they do, viral RNA remains detectable for 3–4 weeks. 

Additionally, a systematic blood sampling of all the animals to conduct serology testing and look for past infection rather than ongoing infection could have helped unravel this bias related to the time windows. In humans, IgG antibodies can be detected at least three months after SARS-CoV-2 infection [44,45,46,47]. However, little is known about the persistence of antibodies in non-human mammals after infection, though the few longitudinal studies available revealed broad interspecific and intraspecific variations in the persistence of IgG antibodies after SARS-CoV-2 [48,49,50]. For example, in captive Malaysian tigers (*Panthera tigris jacksoni*), the antibody response was observed up to 3 months after the first clinical signs of infection [48]. In pets, some cats (*Felis catus*) and dogs (*Canis lupus familiaris*) were seronegative less than three months post-symptoms, while in others, neutralising SARS-CoV-2 antibodies have been detected up to 10 months post-symptoms [49]. In captive white-tailed deer (*Odocoileus virginianus*), the duration of persistence of neutralising antibodies was estimated to be at least 13 months [51]. Notably, some reports mentioning the long persistence of antibody response after SARS-CoV-2 infection in animals have qualified their results, as they were unsure if another infection occurred during their survey [49,50,51]. 

Systematic blood sampling would have involved heavy logistic organisation and animal stress in our case. We therefore relied on the 50 collected serum samples from 26 species that were collected for other purposes, representing about 10% of the mammals from the zoo. Their seronegativity suggests that at least up until 2021, there has been no widespread multi-species SARS-CoV-2 epidemic in the zoos. 

Previously reported cases of SARS-CoV-2 infections in zoo animals have been traced back to asymptomatic infected zookeepers that were in contact with these animals [23,24,25]. The close contact of zookeepers when preparing food, veterinary consultations of animals, or enclosure cleaning represents an important risk of transmission. Since the summer of 2020 and throughout our study, face masks have been worn in the Antwerp and Planckendael Zoos by zookeepers and visitors, in addition to extensive hygiene measures when preparing food and entering the facilities. Likely, the hygiene measures implemented in Antwerp and Planckendael Zoos at the beginning of the pandemic have helped to avoid the transmission of SARS-CoV-2 from humans to animals during most of the pandemic. The origin of the infection of the two hippos at Antwerp Zoo in November 2021 is unknown. The caretakers had no known infection, had no COVID-19 symptoms before the hippo’s infection, and were wearing surgical masks during their work [26]. While several meters of distance is kept from the visitors, as the hippos are housed indoors, aerosol transmission from an infected visitor without perfect masking could have occurred. The genome sequence of the Delta variant with which the hippos were infected was closely related to strains commonly circulating in Belgium at the time [26].

The infection of precisely two hippopotamuses in the Antwerp Zoo was unexpected in the sense that other mammal species have been predicted to be much more susceptible to SARS-CoV-2 based on in silico models of the molecular interaction between the virus Spike protein and the host receptor ACE2 [19,20]. The predictions of SARS-CoV-2′s binding propensity to the hippopotamus viral receptor ACE2 classified the hippopotamus at only medium risk of being infected with SARS-CoV-2, while other taxa, such as primates, were classified as high-risk [19,20]. However, no primates have been reported infected in Antwerp and Planckendael Zoos. The fact that the hippos were housed in an indoor complex where visitors also enter could have contributed to an elevated infection risk for these species. Other species, including bongo, tapir and nutria, that were kept in the vicinity of the infected hippos were negative when sampled right after the reported hippopotamus infections. Visitors did not have access to their indoor enclosure. The hippopotamus infections emphasise that the structural analysis of the SARS-CoV-2 cellular receptor alone is insufficient to estimate the relative spillover risk of SARS-CoV-2 to other animal species [19,20,21]. 

Monitoring zoonotic infections remains the main key to controlling and limiting the spread of zoonotic pathogens. The absence of SARS-CoV-2 in our samples prevented us from expanding the list of potential hosts of SARS-CoV-2. We, however, assessed that SARS-CoV-2 was not circulating in the mammals from Antwerp and Planckendael Zoos between September 2020 and July 2021. The reinforcement of strict hygiene measures and the zoo caretakers’ effective implementation of gloves and masks likely has contributed to avoiding the transmission of SARS-CoV-2 from humans to animals. Therefore, these measures seem efficient in limiting the spread of human pathogens to captive animal populations, and we recommend that these measures be implemented again in case of a new pandemic. 

## Figures and Tables

**Table 1 vetsci-10-00382-t001:** Number of faecal samples collected and tested for SARS-CoV-2 RNA per order, family and species in the two zoos for each sampling session. Session 1: early September 2020, Session 2: mid-October 2020, Session 3: mid-December 2020, Session 4: July 2021, Session 5: December 2021.

			Antwerp						Planckendael					
Order	Family	Species	Session 1	Session 2	Session 3	Session 4	Session 5	Total	Session 1	Session 2	Session 3	Session 4	Session 5	Total
**Cetartiodactyla**	Bovidae	*Addax nasomaculatus*							5	5	5	4		**19**
		*Bison bison*							5	5	5	5		**20**
		*Bison bonasus*							3	3	3	3		**12**
		*Bos taurus*							3	3	3	3		**12**
		*Budorcas taxicolor*	2	2	2	2		**8**						
		*Capra hircus*							5	5	5	5		**20**
		*Cephalophus natalensis*	2	2	2	2		**8**						
		*Gazella leptoceros*							3	3	3	1		**10**
		*Madoqua kirkii*	4	4	4			**12**	5	5	5	3		**18**
		*Nanger dama*							3	3	3	5		**14**
		*Oryx dammah*							1	1	1	1		**4**
		*Oryx leucoryx*							1	1	1	1		**4**
		*Ovis aries*	3	3	3	2		**11**	5	5	5	3		**18**
		*Ovis aries laticaudatus*							2	2	2			**6**
		*Syncerus caffer*	5	5	5	5		**20**						
		*Tragelaphus eurycerus*	3	3	3	3	4	**16**	3	3	3	1		**10**
	Camelidae	*Camelus bactrianus*							5	5	5	5		**20**
		*Lama guanicoe*							5	5	5	3		**18**
		*Vicugna pacos*							5	5	5	10		**25**
		*Vicugna vicugna*							5	5	5	4		**19**
	Cervidae	*Cervus canadensis*							5	5	5	5		**20**
		*Muntiacus reevesi*							5	5	5	5		**20**
	Equidae	*Equus asinus*							2	2	2	2		**8**
		*Equus caballus*							4	4	4			**12**
		*Equus ferus przewalskii*										**4**		**4**
		*Equus grevyi*							5	5	5	5		**20**
		*Equus zebra*	4	4	4	4		**16**						
	Giraffidae	*Giraffa camelopardalis*	3	3	3	3		**12**	5	5	5	5		**20**
		*Okapia johnstoni*	5	5	5	4		**19**						
	Hippopotamidae	*Hippopotamus amphibius*	2	2	2	2		**8**						
	Suidae	*Sus cebifrons*							4	4	4	4		**16**
		*Sus scrofa*							3	3	3	3		**12**
	Tayassuidae	*Catagonus wagneri*							5	5	5	5		**20**
		** *Total* **	**33**	**33**	**33**	**27**	**4**	**130**	**102**	**102**	**102**	**95**		**401**
**Carnivora**	Canidae	*Crocuta crocuta*							2	2	2	3		**9**
		*Speothos venaticus*							5	5	5	5		**20**
	Felidae	*Acinonyx jubatus*							2	2	2	2	2	**10**
		*Panthera leo*	3	3	3	3	6	**18**	3	3	3	3	3	**15**
		*Panthera onca*							1	1	1	1	2	**6**
		*Panthera pardus*							1	1	1			**3**
		*Panthera uncia*							2	2	2	2		**8**
	Herpestidae	*Cynictis penicillata*	5	5	5	5		**20**						
		*Mungos mungo*							5	5	5	5		**20**
		*Suricata suricatta*	5	5	5	3		**18**						
	Mustelidae	*Aonyx cinereus*	1	1	1			**3**	4	4	4			**12**
		*Meles meles*							1	1	1	1		**4**
	Otariidae	*Phoca vitulina*	7	7	7	2		**23**						
		*Zalophus californianus*	4	4	4	2		**14**						
	Procyonidae	*Nasua narica*							1	1	1	2		**5**
		*Nasua nasua*							3	3	3			**9**
		*Procyon lotor*							2	2	2			**6**
	Ursidae	*Ailurus fulgens*							2	2	2	2		**8**
		*Tremarctos ornatus*							2	2	2	2		**8**
		** *Total* **	**25**	**25**	**25**	**15**	**6**	**96**	**36**	**36**	**36**	**28**	**7**	**143**
**Chiroptera**	Pteropodidae	*Rousettus aegyptiacus*							5	5	5	2		**17**
		** *Total* **							**5**	**5**	**5**	**2**		**17**
**Dasyuromorphia**	Dasyuridae	*Sarcophilus harrisii*							3	3	3	3		**12**
		** *Total* **							**3**	**3**	**3**	**3**		**12**
**Diprotodontia**	Macropodidae	*Dendrolagus goodfellowi*	1	1	1			**3**						
		*Macropus giganteus*	4	4	4	3		**15**						
		*Macropus parma*	3	3	3			**9**						
		*Macropus rufus*							1	1		2		**4**
		*Thylogale brunii*	1	1	1			**3**	1	1	1	1		**4**
		*Wallabia bicolor*							5	5	5	5		**20**
	Phascolarctidae	*Phascolarctos cinereus*	1	1	1	2		**5**	2	2	2	1		**7**
	Potoroidae	*Bettongia penicillata*				**1**		**1**						
		** *Total* **	**10**	**10**	**10**	**6**		**36**	**9**	**9**	**8**	**9**		**35**
**Lagomorpha**	Leporidae	*Oryctolagus cuniculus*							5	5	5	5		**20**
		** *Total* **							**5**	**5**	**5**	**5**		**20**
**Macroscelidea**	Macroscelididae	*Rhynchocyon petersi*	3	3	3	2		**11**						
		** *Total* **	**3**	**3**	**3**	**2**		**11**						
**Monotremata**	Tachyglossidae	*Tachyglossus aculeatus*							2	2	2	2		**8**
		** *Total* **							2	2	2	2		8
**Perissodactyla**	Rhinocerotidae	*Ceratotherium simum simum*	2	2	2	2		**8**						
		*Rhinoceros unicornis*							2	2	2	3		**9**
	Tapiridae	*Tapirus indicus*	3	3	3	2	4	**15**						
		** *Total* **	**5**	**5**	**5**	**4**	**4**	**23**	**2**	**2**	**2**	**3**		**9**
**Pilosa**	Myrmecophagidae	*Myrmecophaga tridactyla*							2	2	2	2	2	**10**
		*Tamandua tetradactyla*	1	1	1			**3**						
		** *Total* **	**1**	**1**	**1**			**3**	**2**	**2**	**2**	**2**	**2**	**10**
**Primates**	Aotidae	*Aotus trivirgatus*	1	1	1			**3**						
	Atelidae	*Ateles fusciceps*	5	5	5	5	6	**26**						
	Callitrichidae	*Callimico goeldii*	5	5	5	5	2	**22**						
		*Callithrix geoffroyi*							5	5	5	5	2	**22**
		*Cebuella pygmaea*	1	1	1	1	2	**6**						
		*Leontopithecus chrysomelas*	3	3	3			**9**	4	4	4	3	5	**20**
		*Saguinus imperator*	2	2	2	2	1	**9**						
	Cebidae	*Saimiri boliviensis*							3	3	3			**9**
	Cercopithecidae	*Cercopithecus hamlyni*	3	3	3	4	3	**16**						
		*Colobus guereza*	4	4	4	4	2	**18**						
		*Macaca nigra*							5	5	5	5	3	**23**
		*Macaca sylvanus*							5	5	5	5	3	**23**
		*Mandrillus sphinx*	5	5	5	5	4	**24**						
	Hominidae	*Gorilla beringei*	1	1	1	1	1	**5**						
		*Gorilla gorilla*	5	5	5	5	5	**25**						
		*Pan paniscus*							5	5	5	5	18	**38**
		*Pan troglodytes*	5	5	5	5	11	**31**						
	Hylobatidae	*Nomascus leucogenys*							2	2	2	2	2	**10**
	Lemuridae	*Eulemur macaco*							2	2	2	3	2	**11**
		*Lemur catta*	2	2	2	2		**8**	5	5	5	5	5	**25**
		*Varecia rubra*	1	1	1			**3**						
	Loridae	*Loris lydekkerianus*	5	5	5	5	2	**22**						
	Lorisidae	*Nycticebus pygmaeus*	5	5	5	2		**17**						
		** *Total* **	**53**	**53**	**53**	**46**	**39**	**244**	**36**	**36**	**36**	**33**	**40**	**181**
**Proboscidea**	Elephantidae	*Elephas maximus*	2	2	2			**6**	5	5	5	5		**20**
		** *Total* **	**2**	**2**	**2**			**6**	**5**	**5**	**5**	**5**		**20**
**Rodentia**	Castoridae	*Castor fiber*							1	1	1			**3**
	Caviidae	*Dolichotis patagonum*							5	5	5	5		**20**
		*Hydrochoerus hydrochaeris*							3	3	3	2		**11**
	Dasyproctidae	*Dasyprocta prymnolopha*							1	1	1	1		**4**
	Echimidae	*Myocastor coypus*				5	3	**8**						
	Erethizontidae	*Erethizon dorsatum*	4	4	4	4		**16**						
	Hystricidae	*Hystrix africaeaustralis*				3		**3**	3	3	3	5		**14**
	Murinae	*Lemniscomys barbarus*	5	5	5	5		**20**						
		*Phleomys padillus*	5	5	5	3	1	**19**						
		** *Total* **	**14**	**14**	**14**	**20**	**4**	**66**	**13**	**13**	**13**	**13**		**52**
** *Total* **			146	146	146	120	57	**615**	220	220	219	200	49	**908**

## Data Availability

All data generated or analysed in this study are presented within the tables and figures of the manuscript.

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
