# Peer review of "SARS-CoV-2 Surveillance between 2020 and 2021 of All Mammalian Species in Two Flemish Zoos (Antwerp Zoo and Planckendael Zoo)"

_vetsci, 2023, doi:10.3390/vetsci10060382_

Round 1

Reviewer 1 Report

SARS-CoV-2 surveillance between 2020 and 2021 of all mammalian species in two Flemish zoos (Antwerp Zoo and 3 Planckendael Zoo)

Léa Joffrin et al.

General:

The paper describes the screening of mammalian zoo species for SARS-CoV-2, by faecal and serological screening. None of the investigated samples tested positive for SARS-CoV-2, however, absence of evidence is, of course, not evidence of absence. This is adequately addressed in the discussion, where the frequency of positive faecal samples in proven SARS-CoV-2 cases is discussed and the collection and stability of viral RNA in faecal samples is also discussed.

I would recommend to have a native English speaker make (minor) edits

Reviewer 2 Report

In this manuscript Joffrin et al evaluated the presence of SARS-CoV-2 ( or proof of exposition to the agent through serology) in all mammal species in two zoos in Belgium between September and December 2020 and July 2021. After the detection of SARS-CoV-2 in hippopotamuses in December 31 2021 selected mammal enclosures were studied . A total of 1523 faecal (in pools) samples from 103 mammal species were tested for SARS-CoV-2 via real-time PCR (Pan-CoV system targeting the polymerase gene) and the CDC 2019-nCoV Real-Time RT-PCR Diagnostic Panel specifically targeting SARS-CoV-2 N1 gene. None of the samples tested positive for SARS-CoV-2. Additional surrogate virus neutralization tests were conducted on 50 routinely collected serum samples from 26 mammal species during the same period. These samples were also negative.

As the authors mention, the monitoring of SARS-CoV-2, conducted for several months in all mammal species of two zoos is a plus of their work. In addition, any wildlife information on any viral circulation and specially on the presence of SARS-CoV-2 in potential reservoirs is, today, highly appreciated.

The manuscript is well written, text is clear, techniques are ,in general ,well described and results although negative, are highly valuable considering the type of animals sampled.

Few suggestions mainly regarding the lack of detection of positives

-”A pan-coronavirus system suitable for the detection of alpha-, beta-, gamma- and delta-CoVs real-time PCR adapted version of the Muradrasoli et al. (2009) protocol (32,33) was used on a StepOne™ Real-Time PCR System (Applied Biosystems, Carlsbad, CA) to screen the samples for all potential coronaviruses that may occur in the zoo animals: authors claim that both approaches gave negative results on all samples”

Which positive controls were used to check the pancoronavirus realtime PCR? Since the technique can amplify basically any coronavirus one would imagine that perhaps some other coronavirus aside from SARS-CoV-2, could have been detected (as mentioned by the authors) Please explain

-Were the realtime PCRs checked for the presence of inhibitors in faeces? Please clarify this point since this could be a factor for the lack of positives (realtime PCR may be affected although less than a classical PCR.

Reviewer 3 Report

The manuscript by Joffrin and colleagues presents a good practice for the surveillance of zoo animals for SARS-CoV-2. I think this practice is a good example for the necessity of such studies not only for SARS-CoV-2 but for other pathogens as well. It is also an important example for facilitating the publication of negative results.

Since the manuscript misses important details and have some mistakes or part to be clarified I suggest to conduct a major revision before processing towards the publication.

Major comments:

I really miss the methodological details for the serology tests, these are not included in the supplementary as it is indicated in the MatMethods part of the manuscript. Please include these in the main text, the results may remain as a supplement sincet here is a limitation in brief comm articles.

I dont agree with the last sentence of the abstract and the related discuscussion. This study is not about the monitoring of excretion of SARS-CoV-2 from animals but the exposure of these animals.

Testing the excretion requires totally differenc experimental setup, supplemented with in vitro isolation experiments and some other patrameters. Please carefully revise the whole text in this context. I would also add some discussion and details about the specific actions in this zoo which may have led tot he absence of anti SARS-CoV-2 antibodies in the animals – maybe these details can facilitate some good practice in the future.

Minor comments:

 1. Introduction

47 „Experimental in vivo and in vitro infections showed”  

Please write in vitro an in vivo with italic style

64-47 „Furthermore, the high diversity of animals in zoos, both regarding taxonomy and geographical origin, makes zoos an ideal place to (i) contribute to unraveling the potential host range of SARS-CoV-2 and (ii) evaluate the risk for the conservation of wild animal populations in captivity and in situ.”

Please refer to these sentences in your discussion, whether your study supported this or not.

2. Materials and Methods

96-100 Consider the namind as „Sample preparation, extraction and PCR testing” 

103 Consider to move this table to the supplement  

116-118  „We compared the limit of detection of the Pan-CoV system targeting the polymerase gene to the CDC 2019-nCoV Real-Time RT-PCR Diagnostic 117 Panel specifically targeting SARS-CoV-2 N1 gene”

Please revise the naming of genes, it does not look consistent throughout the manuscript.

124-126 Serological screening: „Serological screening method is detailed in Supplementary material. Details on samples tested are available in Supplementary material Table S1.” 

It is totally missing, please give prompt details in the main text about the method and leave these results in the supplement.

3. Results & Discussion

135 „As such, apart from the infection in two hippos in December 2021 that was discovered because of clinical symptoms and not through our active surveillance study, there was no evidence of SARS-CoV-2 or other coronavirus infection among the mammals residing in the Antwerp and Planckendael Zoos during the time span of the study.” 

Do you connect the possible lack of coronavirus infections (in general) tot he testing strategy (using pan-corona PCR)? I am not sure this statement is valid, since we cannot be sure about the specificity of the test. Please add references or just discuss the SARS-CoV-2 results. 

Another question comes here: was any of the animals vaccinated agains coronaviruses (canine coronavirus for example)?

147-143 It is more related to materials and methods

178-179 „However, little is known about the persistence of antibodies in wild mammals after infection”

Please include studies about the antibody persistance, there is truly a lack from wild animals but there are several from household animals and other model or some susceptible animals (hamsters, mink, etc.).

179-182 It is more related to materials and methods

204-207 Reference is missing here.

218-21 „Monitoring of zoonotic infections remains the main key in controlling and limiting the spread of zoonotic pathogens.”

In zoo setup, where species conservation is also relevant I would not stuck only with zoonotic topic but would describe the importance of protecting these animals. Please revise this last part accordingly.
